# Milestones in the History of Esophagectomy: From Torek to Minimally Invasive Approaches [note 1]

**DOI:** 10.3390/medicina59101786

**Published:** 2023-10-07

**Authors:** Pascal Alexandre Thomas

**Affiliations:** Department of Thoracic Surgery, Lung Transplantation and Diseases of the Esophagus, Aix-Marseille University, Assistance Publique-Hôpitaux de Marseille, North Hospital, Chemin des Bourrely, 13915 Marseille, France; pascalalexandre.thomas@ap-hm.fr; Tel.: +33-491-966-013; Fax: +33-491-966-004

**Keywords:** esophagectomy, history, mechanical staplers, minimally invasive, robot assisted, anastomotic failure

## Abstract

The history of esophagectomy reflects a journey of dedication, collaboration, and technical innovation, with ongoing endeavors aimed at optimizing outcomes and reducing complications. From its early attempts to modern minimally invasive approaches, the journey has been marked by perseverance and innovation. Franz J. A. Torek’s 1913 successful esophageal resection marked a milestone, demonstrating the feasibility of transthoracic esophagectomy and the potential for esophageal cancer cure. However, its high mortality rate posed challenges, and it took almost two decades for similar successes to emerge. Surgical techniques evolved with the left thoracotomy, right thoracotomy, and transhiatal approaches, expanding the indications for resection. Mechanical staplers introduced in the early 20th century transformed anastomosis, reducing complications. The advent of minimally invasive techniques in the 1990s aimed to minimize complications while maintaining oncological efficacy. Robot-assisted esophagectomy further pushed the boundaries of minimally invasive surgery. Collaborative efforts, particularly from the Worldwide Esophageal Cancer Collaboration and the Esophageal Complications Consensus Group, standardized reporting and advanced the understanding of outcomes. The introduction of risk prediction models aids in making informed decisions. Despite significant improvements in survival rates and postoperative mortality, anastomotic leaks remain a concern, with recent rates showing an increase. Prevention strategies include microvascular anastomosis and ischemic preconditioning, yet challenges persist.

## 1. Introduction

“*History is, for every esophageal surgeon, that little cemetery of memories where he goes to seek the lesson of his/her mistakes*”René Leriche (1879–1955)

Esophagectomy has a long and intriguing history. The development of this procedure has been shaped by medical breakthroughs, technological advancements, and the pursuit of improving patient outcomes. From initial attempts to modern minimally invasive techniques, the history of esophagectomy is a testament to the perseverance and ingenuity of surgeons. This article explores the key milestones in the evolution of esophagectomy, shedding light on the crucial references that have guided its progress.

## 2. The Premises: From Franz J. A. Torek (1861–1938) to DeBakey (1908–2008)

Franz John A. Torek was born on 14 April 1861, in Breslau, Germany (now Wroclaw, Poland), a city where both Jan Mikulicz-Radeczki and Ferdinand Sauerbruch worked. His family moved to New York City when he was 11, along with many other Germans seeking refuge from Bismarck’s rule after the German Unification in 1871. After completing his education at the College of the City of New York in 1880, he earned his medical degree from Columbia University’s College of Physicians and Surgeons in 1887. He subsequently became a part of the staff at the German Hospital (now known as Lenox Hill Hospital) in New York City, where he dedicated his entire professional career. On 14 March 1913, he successfully performed an esophageal resection for a subaortic cancer in a 67-year-old female patient through a left thoracotomy [1]. The anesthesia was administered using a mixture of chloroform, ether, and ethyl chloride. Due to the patient’s lack of teeth, an 18F endotracheal intubation catheter was sutured to her upper lip. The general anesthesia was achieved through tracheal insufflation with a Meltzer–Auer apparatus. This device was introduced in 1909 and enabled continuous insufflation of air or oxygen mixed with ether at low pressure using a flexible silk woven catheter inserted through the mouth and larynx into the trachea. This maintained the lungs in a normal distended state, and any excess expiratory air could escape through the trachea around the catheter. The process was facilitated by a foot-bellows that supplied a steady flow of air. The surgical approach was a wide posterolateral thoracotomy through the 6th intercostal space, extended upward by cutting through the 7th, 6th, 5th, and 4th ribs near their tubercles. The esophagectomy was technically demanding, due to intrapleural adhesions, and because the tumor was found fixed under the aortic arch. The left main bronchus was accidentally cut and was subsequently repaired using silk sutures. Pathology disclosed an ulcerated squamous cell carcinoma that was resected in sano. An esophagostomy in the neck and a previously made gastrostomy were joined via an extrathoracic rubber tube, eight days later, allowing oral feeding. The patient refused to have any plastic reconstruction of an antethoracic esophagus but survived for 12 years and eventually died from pneumonia. Torek’s operation marked a major surgical advancement, demonstrating the feasibility of transthoracic esophagectomy and providing evidence that esophageal cancer could be cured by surgery. However, this first success took a long time to be reproduced, as almost twenty years passed between the two, proving the challenge of surviving the operation at that time, as the postoperative mortality rate was reported to exceed 90%. The procedure was thus harshly criticized, with Wolfgang Denk stating, “the result for the patient (when he survives) is deplorable, equivalent to suicide”, and Howard Lilienthal describing Torek’s success as a “disaster for humanity” [2]. In Vienna, 1914, Wolfgang Denk described the same operation on cadavers but without thoracotomy, using a double abdominal and cervical incision, enabling blind dissection, and creating double stomas [3]. Denk himself did not have any clinical success, and in other hands, the procedure was also associated with a similar prohibitively high operative mortality and only 3 patients of the 32 reported survived surgery [4]. In the UK, Georges Grey Turner reintroduced this “pull-through” technique in 1933, with a first successful operation in which esophageal continuity was restored two months later via a presternal channel formed by a skin tube above and a jejunal loop below [5]. In his experience, operative mortality rate was 40%, with only 3 long-term successes among 25 patients, as reported far later by Ogilvie [6]. In 1941, Oschner and DeBakey [4] summarized the outcome, reporting that only 17 of 58 patients undergoing a Torek resection had survived, an operative mortality still exceeding 70%.

## 3. The Challenging Single-Stage Resection and Reconstruction

Initially, the intention was to remove the esophagus. In the rare event that a patient survived, various clever techniques for reconstruction were conceived. However, only a few patients survived and underwent these reconstructive procedures, which, even for those who did, yielded less than satisfactory results. Despite its imperfections, anesthesia, first using positive pressure machines, and later, via intratracheal intubation after the 2nd World War, allowed for more audacious surgeries. Surgeons were not resigned to the discomfort experienced by patients after Torek’s operation. Indeed, double stomas led to an unbearable disability, with a constant discharge of more than one liter of saliva a day at the top, and a poorly fitted orifice for nutrition through a tube, leading to reflux, skin lesions, and tube obstructions or displacements. At best, the patient would not experience hunger. This encouraged surgeons, given the short survival period, and weighing all risks, to attempt in a single intervention the resection and reconstruction by either mobilization of the stomach or the use of jejunum, colon, or skin. The left thoracotomy with phrenotomy facilitated both procedures. There were 8 successes out of 14 cases of such an operation at Imperial University in Kyoto by Tohru Oshawa, as published in 1933 [7]. This was not known to the West until after World War II. Thus, Adams and Phemister’s independent report [8] of a similar operation in 1938 was the one that stimulated emulation in the West (survival for 8 years). In 1945, Richard Sweet, whose name became associated with the left thoracic approach, demonstrated clinically that after division of the left and the short gastric arteries, the stomach was viable and could be mobilized to the arch of the aorta as an esophageal replacement [9]. With the use of the Kocher maneuver to mobilize the duodenum, in 1948, Brewer and Dolley found that the entire stomach could be placed in the chest, making cervical esophagogastric anastomosis feasible [10].

However, this approach still limited esophageal resection to cancers in the lower third. The double abdominal and right thoracic approach marked the emergence of the “royal approach” in the second half of the 1940s and early 50s, which, along with Lewis in the USA [11], Tanner in the UK [12], and Santy in France [13], expanded the indications for resection to tumors in the middle third. Additional methods encompassed the right-sided thoracoabdominal procedure (Ellis technique) involving gastric replacement [14] and utilizing both the small [15] and large [16] intestines as esophageal substitutes. Ronald Belsey’s bon mot “The right approach is the right approach” ruled in most contemporary thoracic surgical theaters. Stomach, colon, and small intestine have been employed as alternatives to the esophagus to create a bypass for cases where esophageal carcinoma was not resectable. This enabled the patient to consume food while concurrently undergoing radiation therapy for cancer treatment. Since the early 20th century, colon interposition has served as an esophageal substitute. The first report of coloplasty dates back to 1911, when Kelling pioneered the procedure [17], and in 1914, Von Hacker achieved the first successful utilization of the colon following an esophagectomy [18]. Since 1945, ascending, transverse, and descending segments of colon, with preservation of their blood supply, have been extensively used for esophageal replacement, Ronald Belsey (Frenchay hospital, Bristol) being the foremost advocate of this technique in the United Kingdom [19]. Paul Orsoni, in France, popularized the use of the left colon as an esophageal substitute in the early 1950s [16], as did Eugène Reboud after he had visited Ronald Belsey in Bristol. Reboud performed numerous retrosternal by-pass coloplasty procedures for patients with unresectable conditions when endoscopic palliation was not yet available [20]. He also advocated colonic interposition as an esophageal substitute after esophagectomy, particularly in children, where preserving the gastric reservoir was crucial for achieving an optimal functional result and preventing nutritional deficiencies [21].

## 4. Advances in Anesthesia and Post-Operative Intensive Care

In the 1940s, the USA saw the appearance of the first closed-circuit anesthesia machines, with CO_2_ absorbers and filters between the mask and the balloon, enabling more audacious surgeries. The end of the war brought solid machines to the market at low prices, originating from US Army surplus, which were easy to handle and continuously improved. After the war, the concept of esophageal teams was emerging, with a chief leading a group consisting of dedicated assistants, nurses, endoscopists, radiologists, and, above all, anesthetists. Garlock emphasized the importance of a competent anesthetist, stating, “without a competent anesthetist, it is better to give up” [2]. The role of an intensivist in post-operative care did not yet exist, and the task was carried out by residents or staff surgeons themselves. The notion of statistics was new, and Garlock emphasized its importance in evaluating results [22]. However, despite the encouraging progress in esophageal surgery, the operative mortality rate remained high, reaching around 25% for lower-third tumors and over 30% for middle-third tumors, even in the best hands. Additionally, only a quarter or a third of operated cancers were resectable. After 1950, the practice of tracheal intubation became routine, although selective bronchial intubation with a Carlens tube [23] would not be introduced for another twenty years. The technique of electric shock for cardiac defibrillation was credited to Beck in 1947 [24]. The use of preserved blood and plasma for transfusion was widely adopted by the American military and allied physicians in 1943 [25]. Curare revolutionized operative conditions but necessitated assisted ventilation. Its use was launched by the Canadian Griffith in 1942 [26]. First-generation antibiotics proved highly effective in treating post-operative infections but were less effective against mediastinal anastomotic fistulas. The complexity of surgery increased with the advancement of assisted ventilation machines. These machines played a crucial role in the anesthetic management and post-operative care of esophageal resections.

## 5. A Technological Revolution: Mechanical Staplers

The manual anastomosis, which was always challenging for a deep and fragile viscera, has also seen technical improvements. The adage “wanting to suture the esophagus is wanting to suture the unsuturable” became outdated. The stitching materials also evolved from silk/catgut and silver/alloy wire of the interbellum years to the synthetic mono, pseudo-mono, and polyfilaments surgical threads offering on absorbable and nonabsorbable forms alike from the early 1970s onwards [27]. The role of the eyeless atraumatic surgical needles unjustly enough, does not receive the credit it deserves in the development of the single plane running sutures and the onset of paradigm shifts in gastro-intestinal anastomoses at around the same time [28]. But it was the advent of surgical staplers that represented a true revolution in the history of esophageal surgery. The first mechanical linear suture clamp was born in the early 20th century in Central Europe, in Budapest, with Hültl and Fischer in 1908 [29]. The heavy (3.5 kg), and difficult-to-load, -sterilize and -handle metal monster was developed further, but the much simpler and more user-friendly Petz machine (1921) gained popularity, mainly in the sphere of influence of German surgery [30]. Later on, the Nakayama stapler (1951), which was robust and allowed surgeons to cut the stomach between two solid rows of staples, proved to be an excellent instrument [31]. Yet, since the staples had to be manually loaded, two or three clamps had to be prepared and sterilized before the surgery. After the 2nd World War, the abdominal surgical staplers based on the Hültl–Petz concept were developed further by the Russian surgeons and their engineers [30]. The subsequent spread of these staplers must be attributed to the wisdom of two American surgeons. In 1958, during the Cold War, these two surgeons, Ravitch (of Russian origin) and Brown, joined an American group for a Soviet congress of hematologists in Kiev to learn about the organization of the blood transfusion service, with a semi-touristic extension to Moscow and Leningrad. In Kiev, they visited the Russian thoracic surgeon Mykola Amosov and incidentally discovered the UKB bronchial suture clamp, realizing that the Russians had a lead of at least ten years in this field. During their stay in Moscow, they visited the Institute of Surgical Instruments and the laboratory of mechanical sutures, where technicians and doctors worked on various disciplines. However, nobody knew or could tell them where they could obtain the UKB clamp. A romantic but inconsistent legend evokes that by chance, in Leningrad, at the terrace of Café Sever, they stumbled upon the right address: the manufacturer of the Red Guard. They went there and purchased the clamp in cash for 440 rubles. Upon their return to Baltimore, Ravitch and Hirsch, a brilliant technical–commercial expert, demonstrated the American efficiency in the following decades. They conducted essential experiments in the laboratory directed by Tim Takato (a Hungarian emigrant), where Félicien Steichen (from Luxemburg) was training. The Russian linear stapler models (UKL, UKB) were improved significantly, as the American derivate implemented ready-to-use sterile reloadable cartridges independent of the device, which made the stapler lighter, and handy. They also designed new devices for different uses of visceral synthesis and anastomosis and engaged in aggressive commercialization by acquiring patents and breaking the initial Soviet–American partnership. The establishment of the US Surgical Corporation and the aggressive and clinically supported marketing led to a quasi-monopolistic position in the market [32]. While the Russians developed end-to-end anastomosis machines (models PKSh, SPTO) for the esophagus and the large bowel, their devices were not reliable enough, mainly for mechatronic reasons [33]. It was not until 1978, when the Auto Suture Company/United Surgical Corporation presented their EEA staplers (entero-enteral anastomosis) offering a safe and elegant surgical machine for the mechanical anastomosis of one of the most delicate reconstructions of tube-like organs of the human body. Regarding anastomoses on the thoracic esophagus, the advent of mechanical sutures resulted in a reduction in the number and severity of anastomotic fistulas, which were the main causes of postoperative mortality. The staplers developed further became lighter, articulated, angled, and downsized as the surgical paradigm shift under the aegis of minimal invasivity/video assistance took place in the operative theaters of the last decade of the 20th century. Thirty years later, the surgical stapler is still indispensable in the robotic surgical procedures of esophageal surgery.

## 6. The Evolution of Surgical Approaches

The evolution of surgical approaches was clearly supported by advancements in anesthesia, intensive care, and instrumental technology. The left thoracic approach, known as the Sweet technique, the first and simplest one, was suitable for resections of lower third esophageal cancers with an esophageal anastomosis below the level of the aortic arch [9]. Lortat-Jacob remained loyal to this approach for cancers in the middle third, requiring good technical mastery for aortic crossing [34]. The right thoracic approach, after an initial abdominal phase of gastrolysis and a change in the patient’s position, provided excellent visibility of the esophagus and the posterior mediastinum. Lewis in the USA [11], Tanner in the UK [12], and Santy in France [13] advocated its advantages as early as 1947, and it became widespread for resections of middle-third and even supra-aortic cancers, leaving sufficient length on the esophagus for an anastomosis with a tubularized stomach. Finally, extending the resection and anastomosis into the neck—the so-called three-hole esophagectomy—was simultaneously described by Kenneth C. McKeown from the UK [35] and Hiroshi Akiyama from Japan [36] in 1976, both techniques differing in the route of the stomach as the esophageal substitute, in the posterior mediastinum, or in the retrosternal space, respectively. The concept of a radical esophagectomy with en bloc resection was initially introduced by Logan in Newcastle, UK [37], and later adopted by Skinner in Chicago, USA [38]. In Japan, extensive two- or three-field lymphadenectomy gained significant attention, with the third field involving a bilateral neck lymph node dissection [39]. Besides these maximally invasive procedures, Mark Orringer in Ann Harbor, USA, in the same period revived the transhiatal esophagectomy to minimize the surgical trauma caused by the thoracotomy, foreshadowing the coming era of minimally invasive approaches, but had to face thunderbolt of comments, including that of Belsey describing transhiatal esophagectomy as an “expedition into the Dark ages” [40]. Nevertheless, whatever the procedures, the worldwide postoperative mortality rate decreased substantially along with time. A comprehensive review of the available literature conducted by Jamieson in 2004 revealed a postoperative mortality of 8% and an overall five-year survival rate of 28% [41].

## 7. The Advent of Minimally Invasive Techniques

The Achilles heel of esophageal cancer surgery has always been its high complication rates, even when performed at high-volume centers, consisting mainly of anastomotic leaks, mediastinitis, and pneumonia. As for other fields of visceral surgery, minimally invasive techniques have been introduced since the early 1990s to minimize this morbidity, without compromising the oncological efficacy of the operation. Alfred Cuschieri from Dundee, UK, is credited with performing the first thoracoscopic thoracic esophagectomy in 1992 [42]. In 1993, Jean Marie Collard in Brussels, Belgium [43], further refined the technique, and in 1995, Aureo Lodovico DePaula in Sao Paulo, Brazil [44], achieved the first completely laparoscopic transhiatal esophagectomy. However, it is Jim Luketich from Pittsburgh, USA, who deserves recognition for developing and popularizing the total minimally invasive esophagectomy (MIE), which is now becoming the standard surgical treatment worldwide [45]. The next decade saw the advent of robot-assisted esophagectomy, which represents the newest innovation in MIE with its own unique benefits and challenges, notably, the need for specific teaching programs and proctored learning, both of which are mandatory. In 2002, Melvin et al. [46] were the first to publish their successful performance of a robotic Ivor Lewis esophagectomy. The following year, in 2003, Horgan reported the first robot-assisted transhiatal esophagectomy [47], and in 2004, Kernstine et al. [48] reported the first fully robotic McKeown three-field esophagectomy. Nowadays, numerous variations of VATS and RATS MIE are performed worldwide, utilizing different combinations of laparoscopic and thoracoscopic approaches as well as hybrid approaches associating various open and endoscopic phases. Along with time, the paradigm of the treatment of esophageal cancer has been changing with the increasing use of minimally invasive esophagectomy in detriment of open esophagectomy. In 2021, a comprehensive systematic review of the available literature was published with a meta-analysis comprising 31 articles, including 5 randomized controlled trials, with a total of 34,465 patients diagnosed with esophageal cancer. It was shown that MIE was feasible even in the context of neoadjuvant treatment and had tendency towards a non-statistically significant decrease in 30- and 90-day mortality. Major cardiovascular and respiratory complications were less frequent in the MIE group. In contrast, no evidence of MIE superiority concerning anastomotic complications was found. Although MIE might contribute to a decrease in minor post-operative complications, MIE was found to be associated with an increased need for a second surgical intervention, and a greater risk for vocal cord lesions; but these results were not statistically significant. Finally, no differences were found concerning risk for local and systemic recurrence, suggesting that MIE did not compromise oncological outcomes [49]. Thus, MIE is just a step forward on a long journey that still has not reached its final destination. Undoubtedly, the quest for an approach that ensures the best oncological results and quality of life, while minimizing postoperative complications, especially anastomotic leaks, remains highly pertinent. At present, the incidence of anastomotic leaks remains in the double digits, underscoring the ongoing significance of addressing this issue.

## 8. The Contemporary Era of Evidence-Based Surgery and International Collaborative Efforts

Besides technical (r)evolutions, most refinements in surgical end-results came from super specialization of surgeons and teams, centralization of esophageal surgery in expert high-volume centers, international collaborations in clinical research, and multicenter prospective controlled studies. Several contemporary surgeons have played a pivotal role in this setting, even if it is impossible to name them all.

Jan van Lanschot, Erasmus University, Rotterdam, the Netherlands, is devoted to the study of esophageal and upper GI cancers and has been working in the field of surgery for over 30 years. In 2002, he led a randomized trial on “Extended transthoracic resection compared with limited transhiatal resection for adenocarcinoma of the esophagus”, which was published in the *New England Journal of Medicine* [50] and concluded that transhiatal esophagectomy was associated with lower morbidity than transthoracic esophagectomy with extended en bloc lymphadenectomy, but a trend toward a worse long-term survival at five years. Also, as the main participant in the ChemoRadiotherapy for Oesophageal cancer followed by Surgery Study (CROSS), he sets up a high reputation in the world establishing a new standard of care for resectable locally advanced cancers of the esophagus and the gastro-esophageal junction [51]. The Netherlands, in general, acts as a leading country in terms of clinical research in the field of esophageal cancer surgery with several teams conducting and/or involved in key studies [50,51,52,53,54,55,56].

Antoon “Toni” Lerut, Leuven, Belgium, has been a registrar in thoracic surgery under the guidance of Ronald Belsey in Bristol, UK. He further honed his skills during a guest scholarship at the University of Chicago, USA, working with distinguished experts David Skinner and Tom DeMeester. In 1976, Lerut returned to the department of general surgery at the Catholic University of Leuven, where he rapidly gained recognition as an exceptional clinical and academic esophageal surgeon, making significant contributions to the field of esophageal cancer through seminal publications [57,58,59,60]. In 1994, as head of general thoracic surgery at KUL, he demonstrated his leadership by assembling a team of bright, young, and academic thoracic surgeons, which culminated in the establishment of the foremost European center for esophageal surgery. Moreover, Lerut is one of the co-founders of the International Society of Diseases of the Esophagus and served as its President from 2001 to 2004, further highlighting his commitment to advancing the knowledge and treatment of esophageal diseases on a global scale.

In France, the first randomized controlled trials about esophagectomy were initiated in the early 1990s by Bernard Launois (Rennes), comparing cervical vs. thoracic anastomosis [61] and transhiatal vs. transthoracic esophagectomy [62]. Much more recently, Christophe Mariette was an example of remarkable expertise in clinical research through his leadership in numerous landmark studies, which have been published in highly prestigious journals, including the MIRO study that demonstrated that hybrid minimally invasive esophagectomy resulted in a lower incidence of intraoperative and postoperative major complications, specifically pulmonary complications, than open esophagectomy, without compromising oncological outcomes [63]. One of his groundbreaking achievements was establishing the ‘FREGAT’ database, a French National database for esophageal and gastric cancer [64]. This initiative paved the way for conducting large population-based studies and set an example for other European countries. Additionally, Mariette played a pivotal role in advancing the modern management of esophago-gastric cancer, particularly in the areas of multimodal treatment. He also emphasized the importance of tailoring treatments to suit the specific needs of individual patient subgroups. Furthermore, his unit at the University Hospital of Lille was a model of dedication to clinical research, and Mariette himself held a prominent position on the international stage.

Thomas W. Rice from the Cleveland Clinic, OH, USA initiated in 2012 the Worldwide Esophageal Cancer Collaboration, which grouped 33 institutions from 6 continents, thereby providing data for more than 22,000 patients with epithelial esophageal cancers. Analytic and consensus processes led to the production of recommendations for clinical and pathological stage groups of esophageal and esophagogastric junction cancer for the AJCC/UICC cancer staging manuals, 8th TNM edition [65,66,67]. This represented a unique opportunity to guide pre-treatment decisions, facilitate prognostication for alternative treatments apart from esophagectomy or endoscopic therapy, and establish a clear and singular therapeutic reference point for esophageal cancer.

Donald Low, based at the Virginia Mason Medical Center in Seattle, USA, is the visionary behind the Esophageal Complications Consensus Group, which had its inaugural meeting in Newcastle, England, in 2011. Utilizing Delphi surveys and face-to-face meetings, this group successfully developed and published a uniform framework for complications and quality measures, along with clear definitions for these complications. This standardized approach has now become the global norm for reporting esophagectomy outcomes on an international scale [68]. As a result of this initiative, the International Esodata Dataset was established, marking the world’s first secure online repository dedicated to recording comprehensive esophagectomy outcomes from across the globe. The Esodata project is actively collecting data from an impressive 75 centers in 21 different countries. Beyond the fundamental accomplishment of standardizing post-esophagectomy outcome reporting, this endeavor also highlighted the immense potential that collaborative international research projects possess. One such project, led by Xavier Benoît D’Journo from Aix-Marseille University in Marseille, France, resulted in the construction of a predictive model of postoperative 90-day mortality based on 10 weighted point variables factored into the prognostic score: age, sex, body mass index, performance status, myocardial infarction, connective tissue disease, peripheral vascular disease, liver disease, neoadjuvant treatment, and hospital volume. The prognostic scores were categorized into five risk groups: very low risk (score, ≥1; 90-day mortality, 1.8%), low risk (score, 0; 90-day mortality, 3.0%), medium risk (score, −1 to −2; 90-day mortality, 5.8%), high risk (score, −3 to −4: 90-day mortality, 8.9%), and very high risk (score, ≤−5; 90-day mortality, 18.2%) [69]. This risk prediction model helps patients and surgeons with assessing the benefit/risk ratio in making informed decisions in daily practice.

Finally, just like in other fields of medicine, Chinese researchers are emerging with the production of high-quality studies, remarkable recruitment capabilities, and short recruitment periods, demonstrating an impressive volume of patients to treat, as typified by the team from Fudan University in Shanghai [70,71,72].

## 9. Provisional Conclusions

Looking back over the shoulder, the history of esophagectomy appears somewhat chaotic. Nevertheless, paraphrasing the French poet Charles Baudelaire in his “Invitation to travel”, “In a journey it is not the destination that counts but always the path taken, and especially the detours”. One century after Torek’s operation, thanks to improved patient selection and the adoption of multimodal approaches, such as induction Chemo +/− Radiotherapy, the overall five-year survival rates have witnessed a notable rise, now ranging from 30% to 40%, even in cases of locally advanced stages. Furthermore, in experienced centers, the 30-day postoperative mortality rate is well below 5%, reflecting significant advancements in treatment outcomes. Obviously, the history of esophagectomy will gain new chapters. One of the pressing challenges that still needs to be addressed is the issue of anastomotic failures. Surprisingly, over the past decade, it seems that the leak rate has experienced an increase once again. According to a recent report from the international Esodata study group, the overall leak rate rose from 11% between 2015 and 2016 to 13% from 2017 to 2018 [73]. This surpasses the 12% rate reported back in the 1980s, and cases with figures as high as 25% are no longer considered uncommon. The rise in anastomotic leak incidence can be attributed to the increased use of induction chemo± radiotherapy, or less precise surgical technique, especially during minimally invasive esophagectomy (MIE) or robot-assisted minimally invasive esophagectomy (RAMIE). It is worth noting that the mortality associated with anastomotic leaks has drastically reduced due to conservative treatments like endoscopic vacuum therapy, clips, and endoprosthesis. However, these treatments can favor the development of tracheo- or bronchoesophageal fistula, which is a devastating complication, particularly when reintervention is considered at a later stage. Consequently, every anastomotic fistula should be regarded as a technical failure. To prevent such complications, ongoing strategies include supercharged microvascular anastomosis and ischemic preconditioning, but their effectiveness remains controversial, thus leaving plenty of scope for technical improvements.

## Data Availability

Not applicable.

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
