# Peer review of "Milestones in the History of Esophagectomy: From Torek to Minimally Invasive Approachesâ€"

_medicina, 2023, doi:10.3390/medicina59101786_

Round 1
Reviewer 1 Report
Dear author,
First I congratulate you for your successful review article entitled ' Milestones in the history of esophagectomy: from Torek to minimally invasive approaches'.
However I have some comments.
1)It is advisable to add your own experiences and comments on the subject throughout the text.
2)Please attach a table summarising the history of esophagectomy.
3)You can include some pictures of your own experience of esophagectomy and the history of esophagectomy to increase the reader's interest.
Thank you.
Minor editing of English language required
Author Response
I appreciate your comments. To stick to the instructions of the guest editor of the special issue in which this paper will be published, I did not refer at all to my personal experience and focused the draft on historical milestones. However, the contribution of my team in the field is reflected in the references list. Besides, well-known historical illustrations (i.e., Torek's patient, first staplers, etc..) are already available in several publications, and thus were not considered for copyright issues. With my best regards. P. Thomas
Reviewer 2 Report
This paper covers the history of esophagectomy. It is fluently written and a fascinating read. Every surgeon with just a faint historical interest will profit from this paper.
This work bases its strength on the history of the earlier esophagectomies. It is difficult to adequately depict the actual, modern time. So, in my opinion, I’d concentrate on these times and omit the names of modern surgeons, just mentioning the direction the development takes.
Further, if ever possible, I think a few illustrations/pictures would improve the paper even more.
Author Response
Thank you for your comments and suggestions. To stick to the instructions I received from the guest editor of the special issue of the journal in which the paper will be published, the period covered by this historical perspective had to include the "modern times". This is why the works of contemporary surgeons were detailed. Overall, illustrations were omitted to save space given the unusual length of the manuscript, and for copyright issues. Best regards.